

# Critical reflections on three popular computational linguistic approaches to examine Twitter discourses

Dan Heaton, Jeremie Clos, Elena Nichele and Joel Fischer

School of Computer Science, University of Nottingham, Nottingham, United Kingdom

## ABSTRACT

Although computational linguistic methods—such as topic modelling, sentiment analysis and emotion detection—can provide social media researchers with insights into online public discourses, it is not inherent as to how these methods should be used, with a lack of transparent instructions on how to apply them in a critical way. There is a growing body of work focusing on the strengths and shortcomings of these methods. Through applying best practices for using these methods within the literature, we focus on setting expectations, presenting trajectories, examining with context and critically reflecting on the diachronic Twitter discourse of two case studies: the longitudinal discourse of the NHS Covid-19 digital contact-tracing app and the snapshot discourse of the Ofqual A Level grade calculation algorithm, both related to the UK. We identified difficulties in interpretation and potential application in all three of the approaches. Other shortcomings, such the detection of negation and sarcasm, were also found. We discuss the need for further transparency of these methods for diachronic social media researchers, including the potential for combining these approaches with qualitative ones—such as corpus linguistics and critical discourse analysis—in a more formal framework.

## INTRODUCTION

Computational linguistics lend themselves to study public discourses around topics of significant current societal concern. With most cultural phenomena, social media websites such as Twitter host a plethora of views relating to current affairs, and there is interest in gathering these views to make judgements on the general trends of opinion expressed (*McCormick et al., 2017*). Many offer opinions on this public-access site, providing a large dataset that can be analysed by means of open APIs in real-time (*Kumar, Morstatter & Liu, 2014*). A popular way to explore the views expressed on social media is to use 'off-the-shelf' computational linguistic methods such as topic modelling, sentiment analysis and emotion detection. This is because these methods may be less intrusive and more cost-effective when compared with interviews or experiments (*Rout et al., 2018*).

Social media researchers have used methods such as topic modelling, sentiment analysis and emotion detection to mine social media and online spaces for views expressed about

Corresponding author
Dan Heaton,
daniel.heaton@nottingham.ac.uk

various topics. Cases of interest include research by *Hu, Chancellor & De Choudhury (2019)* into discourses relating to homelessness on social media, where topic modelling techniques were deployed to investigate common thematic threads between those who identify as homeless and those who do not. Sentiment analysis and emotion detection techniques have been explored by *Tang et al. (2020)* and *Wang et al. (2021)*, who both used them to garner greater insights into discourses relating to online and remote education.

However, there is now a growing body of work involving topic modelling, sentiment analysis and emotion detection that focuses on critically examining how these methods can be applied and the relative success of the methods. There have been studies that focus on the existing limited critical examination of particularly the shortcomings of these methods when applied to large corpora drawn from social media. Some initial evaluative work has been done in the studies by *Jiang, Brubaker & Fiesler (2017)*, who critiqued the binary sentiment scale used in sentiment analysis packages as having little tangible meaning about the views expressed in text. Additionally, emotion modelling critiques by social media researchers include work by *Fast, Chen & Bernstein (2016)* and their comparison of numerous models to develop a better performing function.

Therefore, to further contribute to this discourse, we perform an exploratory critical reflection in this article. We do this by following an analysis process that draws on best practices. We then draw out instances from our analysis of Twitter data that demonstrate the application of these methods and how they have been used to capture meaning from the data. By using the critically reflective weather model by *Maclean (2016)* as the basis for this, we look at how this could aid social media researchers in the use of these methods.

Two examples of public discourses around topics of societal concern have been chosen to aid this critical reflection. The first, the Covid-19 contact-tracing application, was launched in the UK with the intention to easily identify users who have come into contact with someone who has tested positive for Covid-19, based on Bluetooth proximity technology. The app has had widespread impact on the lives of citizens of the country such as by informing them to self-isolate at home for up to 10 days (*Kretzschmar et al., 2020*). It is estimated that at the height of the recent wave the app has sent more than 600,000 self-isolation alerts in a single week in July 2021 alone (https://www.bbc.co.uk/news/technology-57929162). Related to our research, Twitter was one of the websites used to explore attitudes to digital contact-tracing in Germany, where issues of privacy were among the most intensely discussed topics (*Arzt, Poller & Vallejo, 2021*). The second example, the Ofqual algorithm used for calculating A Level grades in England and Wales in 2020, was initially brought in to replace examinations in the Covid-19 pandemic. However, public outcry saw the algorithm scrapped and teacher assessment grades used instead. These examples were chosen due to having contrasting timelines (an 11 month period for the COVID-19 app and a three week period for the Ofqual algorithm).

The aims of this article is to critically reflect on the use of topic modelling, sentiment analysis and emotion detection when analysing social media data. We do this *via* following a set of collated best practices from within the field. Our work does not aim to provide an exhaustive analysis of the two case studies on social media more broadly, but to examine two focused datasets from a singular source (Twitter) to offer critical reflections on the

application of computational linguistic methods. As a result of this, we offer a set of reflections to be used by social media researchers when undertaking further inquiries into online discourses that could benefit from using these methods.

# RELATED WORK

Herein, we introduce the computational methods we chose to examine our Twitter datasets, review related work in the realm of social media analysis and consider existing approaches to addressing shortcomings in these approaches. This review will uncover existing best practices and motivate how we combine these to create our analytical process. We also provide a brief overview of the digital contact tracing in the UK and the Ofqual A Level algorithm for context.

## Computational linguistic methods

Three popular computational linguistic tools have been chosen as the means the explore two public discourse case studies on Twitter. The three approaches—topic modelling, sentiment analysis and emotion detection—are united in their use of language to either describe or make predictions about a corpus. Thus, they are descriptive and predictive in their function. They are furthermore popular choices when there is a large amount of linguistic data to explore. Topic modelling is best used to uncover latent topics present within large bodies of text (*Nikolenko, Koltcov & Koltsova, 2017*). Sentiment analysis—otherwise known as opinion mining—uses predictive algorithms on a binary polarity scale to provide insight into the views expressed in text (*Liu, 2010*; *Vyas & Uma, 2018*). Emotion detection methods use similar predictive detection algorithms to sentiment analysis to ascertain emotions or states of being that may be expressed within a text (*Sailunaz et al., 2018*).

We are also aware of a number of other techniques that could have been used, but have chosen to focus on these three due to scope, suitability and accessibility. Ultimately, these three chosen methods have consistently been used by researchers within this realm of social media discourses. They have also been chosen as their scores can be compared in a diachronic way.

### Topic modelling

Automated topic modelling, particularly Latent Dirichlet Allocation (LDA), is seen as beneficial in qualitative text studies due to its focus on uncovering underlying topics present within a series of documents (*Nikolenko, Koltcov & Koltsova, 2017*). The use of LDA as a topic modelling method with Twitter data has grown in interest in recent years (*Arianto & Anuraga, 2020*) and various techniques have been designed to undertake this investigation. There has been focus on using bigrams—pairs of adjacent words—to form topic models through LDA when investigating views expressed on Twitter (*B.Srinivasan & Kumar, 2019*). This is closely related to the work done by *Yang & Zhang (2018)*, who combined this with the Bag-of-Words (BoW) model. This aided the creation of reliable topic models when working with Twitter data, especially when concerned with context dependence of short texts. *Prihatini et al. (2018)* combined, compared and contrasted LDA with Term

Frequency Inverse Document Frequency (TF-IDF), a method regularly used for feature extraction of texts, when examining online news and related media articles. They found that the Precision, Recall and F-Measure values of LDA was higher than TF-IDF for predicting topics, thus recommending it as the more suitable method.

The gensim package is a popular choice for topic modelling and LDA, using the analysis of co-occurrence patterns to identify latent structures in plain text documents (*Rehùøek & Sojka, 2011*). *Hidayatullah et al. (2019)* used gensim as the LDA model to investigate topics and trends regarding climate change and weather on Twitter, where five key topics were able to be defined through this method. Another relevant study was by *Song et al. (2019)*, who investigated topics in media discourses regarding illegal compensation given to victims of occupational injuries. The topics discovered were then used as recommendation points for ensuring efficient occupational health and safety schemes protect vulnerable employees from illegal practices, exemplifying the practical applications that using LDA can have.

This method has been been used within contemporary social media studies. *Hu, Chancellor & De Choudhury (2019)* used gensim as their topic modelling tool when investigating discourses relating to homelessness on social media. They looked first at the the blog posts of those who identify as homeless on the blogging site tumblr and compared these to the blog posts of those who do not identify as homeless. By using LDA, thirteen latent topics were identified as part of the discourse of homeless blog posts and seventeen topics were uncovered to be part of the control group's discourse. Within this study, LDA was deemed successful in its identification of the different ways in which homeless people and non-homeless people discuss the topic of homelessness. It was recommended that organisations can tap into these topical discourses to raise awareness and promote support, exemplifying the power of using this computational linguistic method. Additionally, *Sengupta (2019)* also used LDA to investigate latent topics in subreddit forums. This combined manual inspection with LDA to validate findings.

It has been documented about how to best find the appropriate number of topics for a dataset when using LDA. In their exploration of NLP-based techniques, *Nguyen et al. (2020)* suggested that the needs of the researcher must be examined: a small number of topics for a broad overview, and more topics for finer detail.

There have been critiques of topic modelling also. *Maier et al. (2018)* questioned the validity and reliability of LDA and offered an evaluative framework that enabled communication researchers to more effectively deploy this method. Their four recommendations were categorised as pre-processing the data, selecting parameters carefully, evaluating the reliability of the model and, finally, checking the validation of the results *via* manual review. We aim to extend this idea to other computational linguistic methods *via* our own examination and reflection. Researchers have also explored the limitations of 'off-the-shelf' topic modelling. In particular, stemmers (which are used to conflate several words to a shared meaning in topic modelling) were critiqued by *Schofield & Mimno (2016)*, suggesting that they had no effect on the outcomes of topic models and often disrupted topic stability. Contemporary Twitter studies such as *Saura, Ribeiro-Soriano & Saldaña (2022)* also expressed that labelling LDA topics is a manual process and, as such, this introduced bias into their results.

### Sentiment analysis

Another popular investigation method to uncover the views expressed on social media is sentiment analysis. Sentiment analysis is defined as 'the computational treatment of opinions, sentiments and subjectivity of text' (*Medhat, Hassan & Korashy, 2014*). In its common form, it uses a binary polarity scale from negative to positive, with neutral in between, initially examining lexicon individually with a view to providing an overall sentiment of a text (*Liu, 2010*). It is often used to easily ascertain data that will provide insight into the opinions of others (*Vyas & Uma, 2018*), which enables investigation through a predictive element. Sentiment analysis has been used to investigate meaning of Twitter discourses for this same reason (*Liu & Zhang, 2012*). For this particular study, we chose to focus on dictionary-based approaches as these were the most common approaches seen in similar studies. Nevertheless, there are other approaches, such as CNN or deep learning methods, that are becoming popular within NLP.

A staple of most dictionary-based sentiment analysis methods is the Naive Bayes classifier, which makes the simplifying assumption that all words are sampled independently (hence the 'naive'). This produces a conditional probability of a document belonging to a category (*Rish et al., 2001*). *Pak & Paroubek (2010)* built a classifier based on Naive Bayes principles that classified Twitter data about a range of topics according to traditional sentiment polarity, but claimed it outperformed other comparable classifiers in accuracy. *Ulfa et al. (2018)* combined the Naive Bayes classifier with a Mutual Information method when examining tweets relating to tourism in Lombok and yielded a high classification accuracy.

Two popular sentiment analysis modules that have been used for Twitter opinion mining are TextBlob and VADER. TextBlob also uses the Naives Bayes model and also provides a subjectivity measure (*Loria, 2018*). *Pokharel (2020)* used this technique to analyse the response to the COVID-19 outbreak in Nepal and found that majority of the people of Nepal were taking a positive and hopeful approach, but there were instances of fear, sadness and disgust exhibited too. Additionally, *Sivalakshmi et al. (2021)* explored the sentiment towards the COVID-19 vaccine using TextBlob and concluded that the discourse was neutral-to-negative in polarity. An important factor here was that they identified that TextBlob was unable to read tokenized special characters as a limitation of the module, and factored this into their analysis. *Tang et al. (2020)* adopted the TextBlob sentiment software within their ConceptGuide system. The sentiment analysis here played a crucial element in the evaluation of their tool and future work proposes a learning efficiency analysis that may use similar principles, exemplifying the utilisation of TextBlob outside of a social media context.

VADER classification module acts in a similar way to TextBlob, using bigrams to attempt to detect negation in syntactical structures. Various studies have ranged from mining emotions from online video comments (*Chaithra, 2019*) to looking specifically at sentiment on Twitter. Chauhan et al. recognised the changing and challenging formats of tweets might have had an impact on previous work undertaken, and, because of VADER's sensitivity to social media formats, it is a more suitable module to use and yielded more accurate results (*Chauhan, Bansal & Goel, 2018*). *Mustaqim et al. (2020)* combined VADER's with

the k-nearest neighbour algorithm, and found that the two of them together yielded greater insight into the Twitter discourse concerning Indonesian forest fires in 2019. Because of its suitability for social media research, VADER was used by *Park, Ciampaglia & Ferrara (2016)* to investigate fashion trends on Instagram. Additionally, VADER has applications beyond social media discourse research. It was used in the study about student perceptions of a virtual teaching assistant in a study by *Wang et al. (2021)*.

*Gupta & Joshi (2021)* looked specifically at the role of negation in Twitter sentiment analysis, with a focus is on negation scope detection and negation handling methods. Their main conclusion is that negation is not a trivial task but entails many challenges, such as implicit negations and negation exception cases. By exploring this within the healthcare domain on Twitter, they have been able to lay foundations for this to be applied to other areas of research on Twitter, stressing the importance of handling negation exception cases, where negation cue act as non-cue, through a deep learning model. In similar fashion, sentiment analysis modules have been critiqued for the lack of consideration for sarcasm. *González-Ibánez, Muresan & Wacholder (2011)* investigated this by comparing machine learning techniques to human reviewers, and found that neither performed well in classifying sarcastic tweets. *Villena-Román & Garcıa-Morera (2013)* built on this when suggesting that sentiment analysis is so complex that humans will often disagree on the sentiment of a text in question.

There have been developments to the standard sentiment analysis models in recent years, with social media researchers going beyond the standard approach. For example, *Watanabe (2021)* set forth an alternative to sentiment analysis entitled Latent Semantic Scaling (LSS), which used principles of latent semantic analysis (word embeddings) to improve the traditional sentiment model. In addition, *van Atteveldt, van der Velden & Boukes (2021)* deployed a survey of many different sentiment analysis techniques and concluded that sentiment dictionaries were not of acceptable standards of validity and, while machine and deep learning outperformed dictionary-based methods the best performance was attained by trained human coding. This further shows that work to combine quantitative and qualitative methods here may be of use. All the studies exemplify that, to make best use of these computational language analysis methods, they must be combined with other approaches to validate results.

Some studies have begun to offer ideas about how to overcome the limitations of sentiment analysis. Some of these have been technical, such as *Ribeiro et al. (2020)* evaluating the number of actionable bugs through using an agnostic testing methodology for NLP models. However, some have focused on interpretation. For example. the study by *Agarwal et al. (2015)* offered the idea of using contextual information alongside sentiment results to better interpret them. This goes alongside other suggestions, such as the research into presenting sentiment over a period of time as a trajectory by *Howard (2021)*. This allows researchers to see how sentiment trends begin to form and develop, thus offering sentiment change as an interpretation strand. These ideas were important to us when considering our approach to analysis.

### Emotion detection

Emotion detection from text could be seen as a sister method to sentiment analysis in the sense that it attempts to assign to documents a multidimensional vector representing its emotional valence (resp. sentiment valence) across a set of pre-defined emotion categories (resp. sentiment polarity), based on observation of text. One of the earliest pieces of work from the profile of mood states set forth by *Bollen, Mao & Pepe (2011)*. They used a psychometric instrument to extract six mood states (tension, depression, anger, vigor, fatigue, confusion) from the aggregated Twitter content and found that social, political, cultural and economic events are correlated with significant, even if delayed fluctuations of mood.

A popular contemporary emotion detection module for Python is EmoLex. The algorithm takes a list of English words and their associations with eight basic emotions (anger, fear, anticipation, trust, surprise, sadness, joy, and disgust), which was manually done initially by crowdsourcing (*Mohammad & Turney, 2013*). This model has been applied to Twitter investigations, such as the analysis by *Aribowo & Khomsah (2021)* of Indonesian Twitter users' response to the COVID-19 pandemic. *Mathur, Kubde & Vaidya (2020)* again used a similar process to find high levels of trust and fear in tweets relating to COVID-19 from all over the world. Facebook data has also been investigated using EmoLex, with notable examples being the two 2019 studies by Balakrishnan et al. that investigated the emotions in the online diabetes community. EmoLex was refined using string-based Multinomial Naïve Bayes algorithm, with results indicating a 6.3% improvement (*i.e.,* 82% *vs.* 75.7% for average F-score) when compared to the EmoLex alone (*Balakrishnan et al., 2019*; *Balakrishnan & Kaur, 2019*).

The examples here illustrate the benefits of using EmoLex and emotion detection models within a social media context. However, within most of these studies in a social media context, there has been limited focus on the shortcomings of using these methods. Some studies have commented on the need for more complex emotional categories (*Jiang, Brubaker & Fiesler, 2017*), which may solve overlap between categories (*De Silva et al., 2018*; *Leung et al., 2021*). Also, some have attempted to mitigate human review biases seen when classifying texts as 'neutral' because of annotator's uncertainty around the best-fit category, rather than it actually being neutral (*Fujioka et al., 2019*). Additionally, EmoLex was used by *Fast, Chen & Bernstein (2016)* to develop Empath, a tool that can generate and validate new lexical categories on demand from a small set of seed terms to use of large texts. This offered some critique of EmoLex, as they stated that it correlates imperfectly with Linguistic Inquiry and Word Count analytical procedures. This is an important starting point for a more open, critically reflective discussion on the practical advantages and limitations of using each of these computational linguistic methods when investigating social media discourses. A reflection on the use of these methods may benefit those in the future that are looking to deploy them as part of their investigation.

### Summary and research gap

Overall, this review shows that these computational methods are used regularly in the social media research community, although exploring them critically through the medium

of reflection is a newer discourse. While there are studies that have used these methods, or similar ones or variations, alongside qualitative methods, few have been formalised so that they include existing best practices and critically reflective ideas. Therefore, this contribution will combine existing best practice ideas through an analytical process that encourages critical reflections.

## Context of public discourse case studies

We chose two recent case studies of public discourses as examples for our work. These two—the NHS COVID-19 digital contact tracing app and the Ofqual A Level algorithm—were chosen due to their significant societal concern in the United Kingdom and their contrasting timelines (a dataset spanning eleven months for the COVID-19 app and a dataset spanning three weeks for the Ofqual algorithm). This section introduces some background context of these two case studies.

### Digital contact tracing during COVID-19 in the UK

NHS COVID-19, the contact-tracing application created by Serco on behalf of the UK government to track active cases of COVID-19, has impacted daily life in the United Kingdom since its launch (*Kretzschmar et al., 2020*). The application is available on mobile phones and uses exposure logging, developed by Apple and Google (*NHS England, 2021*). This technology allows the application to send alerts, using a randomly generated identification number, when the user is near to another application user who has logged a positive COVID-19 test. Some of the issues users encountered were backwards incompatibility, incorrect alerts and false positive tests, which meant that users had to self-isolate for ten days when the result was incorrect, impacting income and well-being (*Kent, 2020*). *Mbwogge (2021)* claims that a symptom-based contact-tracing system failed to meet the testing and tracing needs in the United Kingdom, which is further evidenced by the fact that cases of and deaths relating to COVID-19 increased to be the highest in Europe. Although adoption of the application was encouraged, the uptake of the application was less than expected at 20.9 million downloads, with no specific figures published on active users.

There are a growing number of research projects investigating the public attitudes towards digital contact-tracing in the UK. *Williams et al. (2021)* interviewed 27 participants over online video conferencing before the release of the COVID-19 app in the UK and found the response to be mixed and heavily influenced by moral reasoning. Analysis revealed five themes: lack of information and misconceptions surrounding COVID-19 contact-tracing apps; concerns over privacy; concerns over stigma; concerns over uptake; and contact-tracing as the 'greater good'. *Samuel et al. (2021)* conducted 35 semi-structured qualitative interviews in April 2020, showing interviewees' views about the potential of app for contact tracing. Participants showcased a range of misconceptions and worries. However, as there was no follow-up to this study, it was impossible to discover which of the participants would then choose to download or not to download the app once it was launched in September 2020.

After the app launched, *Dowthwaite et al. (2021)* found through surveying 1,001 UK adults that half of participants installed the app and a high number of these participants

complied with the app on a regular basis. They also found that there were issues surrounding trust and understanding that hindered the effective adoption of the app.

In July 2021, media focus was placed on the app once more due to the amount of alerts to self-isolate being sent to users. An increased amount of positive COVID-19 cases in the UK and the relaxation of government restrictions meant that users of the app were more frequently contacted due to exposure logging (*Abbasi, 2021*). This was negative perceived in the press and, as a result, the pejorative portmanteau 'pingdemic' was coined (*Rimmer, 2021*).

### The Ofqual A level algorithm in England and Wales

Ofqual's use of an algorithm for calculating Advanced Level qualification grades in the summer of 2020 was a highly contested issue (*Kelly, 2021*). The algorithm was deployed to replace the standard A Level qualification, which had been abandoned due to the COVID-19 pandemic. Instead, the algorithm used data such as prior centre attainment and postcodes and factored these into the generated grade for each qualification. This led to a number of students receiving grades that were different to their predicted outcomes submitted by their teachers when the results were released to students on August 13th 2020.

Despite defending the use of the algorithm initially, due to public outcry, the UK Government retracted the use of the algorithm-decided grades on August 17th 2020 and all qualifications were awarded the teacher-submitted grade instead (*BBC, 2020*). The public reaction also saw the resignations of Sally Collier, CEO and Chief Regulator and Ofqual, and Jonathan Slater, the most senior civil servant in the Department for Education.

A report published stated no grading bias  (*Ofqual, 2020*) but it was found that the algorithm favoured students from more economically privileged backgrounds and meant that students from less economically privileged backgrounds suffered the most (*Smith, 2020*). This led to it being labelled a 'mutant algorithm' by UK Prime Minister Boris Johnson (*Coughlan, 2020*).

## METHOD

### Data collection and processing

Data collection occurred through using the Twitter for Academic Purposes Application Programming Interface. Using Twitter as a data source here is significant due to the availability of the data, the amount of data and the ability to analyse it in real time. As stated our aim wasn't to conduct a comprehensive analysis of the discourse on social media more broadly, but to examine a specific dataset to critically reflect on computational linguistic methods. Using Tweets as a specific social media dataset comes with many risks and rewards (*Agarwal et al., 2011*). The fact that data can be collected from Twitter in real-time supports the current computational linguistic analysis models that have the functionality to do the same (*Kumar, Morstatter & Liu, 2014*). An important aspect of using Twitter is that its data can be pre-processed before analysis (*Jianqiang, 2015*) and lends itself well to exploratory analysis principles (*Chong, Selvaretnam & Soon, 2014*).

There are some complex ethical considerations when scraping data from Twitter for data analysis. A prominent ethical issue is the fact that although tweets are public(by default), Twitter 'data' isn't provided by users for the purposes of research, and that gaining explicit consent to use their tweets for research is practically infeasible (*Woodfield et al., 2013*). We followed best practices recommended in social media research literature, including not including any screenshots of tweets that may be later identifiable without first gaining consent from the tweet author. Tweets were anonymised to researchers as part of the data cleaning process. This study was approved by our university department's ethics committee.

Data extraction occurred using the Tweepy module in the Python programming language (*Roesslein, 2009*). For the COVID-19 app, the key search criteria for this was tweets containing '@NHSCOVID19app', which is the official Twitter handle for the UK's contact-tracing app, and the related hashtag '#NHSCOVID19app'. The reason for this choice was to ensure that tweets were directly related to experience of the contact-tracing app itself, rather than the NHS Test and Trace system as a whole. Although key parts of the discourse may not be revealed through this search term alone, it provides a starting point for investigating views expressed about the app. In total, 180,281 tweets (1,797,052 words) were taken from 23rd September 2020, the day before the app launched in the UK, to 31st July 2021. After this, tweets were condensed down to remove advertisements from the dataset, resulting in a final dataset of 82,294 tweets over an eleven month period. For the Ofqual algorithm, the search criteria was tweets containing any of the following: 'Ofqual algorithm', 'ofqualalgorithm', 'A level algorithm', 'alevelalgorithm', 'a levels algorithm', 'a-level algorithm' or 'a-levels algorithm'. In total, 18,239 tweets (509,948 words) were taken from 12th August 2020, the day before A Level results were released to students, until 3rd September 2020, after Ofqual's chair appeared at the Education Select Committee. The data was sourced from the United Kingdom and only tweets in English were selected. Therefore, the analysis investigated views expressed in English only.

Data was pseudonymised during extraction, with a unique number generated to refer to each tweet. Stopwords were removed from the dataset using gensim, along with the removal of all long and short URLs and the indication 'RT' (retweet) at the beginning of any tweet. Twitter handles that appeared within tweets were also redacted, using gensim, for anonymity.

## Computational linguistic methods
### Topic modelling
The topic modelling technique chosen was LDA and the gensim module was chosen to perform this. The existing data was tokenised using gensim's 'simple pre-process' function and bigram and trigram models were created using the 'phrases' function. The process also allowed bigrams to be made and lemmatised using Natural Language Toolkit (*Bird, Klein & Loper, 2009*). The id2word dictionary was inputted and combined with the gensim corpora to create the dictionary specific to this dataset. Each word in the document was given a unique ID. Then, using the dictionary, the corpus, which was a mapping of the word ID and the word frequency, was created (*Rehùøek & Sojka, 2011*). The topics were

then generated and printed using the 'gensim.models.ldamodel.LdaModel' function within gensim.

### Sentiment analysis

Two different sentiment analysis approaches were chosen—one that used TextBlob and the other using VADER. This was done to make comparisons between the two systems—especially with the overt claim that VADER accounts for negation within its algorithm. The TextBlob model was programmed with a training set about COVID-19 (*Lamsal, 2021*) to become familiar with COVID-related lexicon using the 'train' and 'test' commands. We did not use this for VADER as it is a pre-trained analyser. For TextBlob, a CSV file containing tweets only was imported into the python library and the command 'blob = TextBlob(sentence)' executed. For VADER, the function was 'sentiment_analyzer_score' and parameters were set so that each tweet could labelled 'positive', 'negative' or 'neutral'—with 0.05 and above being 'positive' and −0.05 and below being 'negative'.

### Emotion detection

The emotion detection module run was EmoLex, also in Python. The 'top.emotions' command was deployed, Which exported a CSV table with each tweet's correlation to each emotion—fear, anger, anticipation, trust, surprise, sadness, disgust and joy—declared, along with labelling the dominant emotion in a tweet in a separate column.

## Analysis process

To formalise our process, we searched for a framework that would enable us to engage with the methods and critically reflect on our use of them in each of the case studies. However, this search was unsuccessful—while many offered frameworks that were specific to the methods being used, missing elements included how to frame critical reflection within this and ensure its universal relevance to our three chosen methods. As a result of not being able to find a well-defined whole model, we have combined other existing recommendations for best practice in a process that could be replicated by other social media researchers.

The five steps in this process helped guide us through the process of using each of the methods. This is a combination of borrowed best practices and enables critical reflection through the use of Maclean's model. Once the method of analysis is chosen, depending on what is being examined and the aim of the research (*Heitmann et al., 2020*), the steps we followed were:

1. **Set expectations:** record what you hope to find in the discourse from using computational linguistic methods. Setting expectations is advocated by *Post, Visser & Buis (2017)*, who suggest that, by writing down expectations prior to the start of the data collection and analysis, the reflection after this is complete will be much more fruitful.

2. **View as trajectories:** present data chronologically to show which topics are discussed, the sentiment of views expressed or the emotions detected. This is a good place to begin to see patterns and areas of interest in the data. Presenting longitudinal data as a trajectory is advocated by *Howard (2021)* and compliments how trends can be seen quickly trhough real-time data collection (*Alamoodi et al., 2021*).

3. **Human review:** according to similar studies (*González-Ibánez, Muresan & Wacholder, 2011*; *Maier et al., 2018*), it is important to human review a sample of the tweets. This offers us the opportunity to not only classify the tweets according to the categories defined by each tool, but also annotate instances of potential inaccuracy, such as sarcasm or negation. The human review was undertaken by two different reviewers and inter-annotator agreement calculated.

4. **Examine items of interest with context:** whether they are turning points, extreme polarities or suggest they have been questionably categorised, examining these with contextual data, such as knowledge about events that move the public at the time, may help create more meaning from the results, as per the suggestions of *Agarwal et al. (2015)*.

5. **Conduct formal critical reflection:** formally conduct critical reflection using Maclean's weather model. Use the expectations recorded before using the method to measure its success and suitability for analysis on this occasion.

## Critical reflection model

After exploring each of the methods using the process, a critical reflection takes place. *Finlay (2008)* defines critical reflection as 'learning through and from experience towards gaining new insights of self and practice'. Critical reflection is not a new concept within NLP, HCI and social media research (*Sengers, McCarthy & Dourish, 2006*) but a greater focus has been placed in the design process of new or developing technology. Within our work, we employ critical reflection to examine how suitable the method has been to investigate the public discourse surrounding digital contact-tracing in the UK by applying a simple four strand critically reflective model outlined by *Maclean (2016)*. Using principles from other models, such as the one by *Gibbs & Unit (1988)*, Maclean provides the following stages:

- Sunshine—what went well?
- Rain—what didn't go well?
- Lightning—what came as a shock or surprise?
- Fog—what wasn't understood or could be a further challenge?

Although initially used for educational practitioner reflection, this model allows for simple yet robust critical reflections that we thought would provide a concise format to present lessons learnt in an accessible form for other social media researchers like us who are not experts in computational linguistics. Maclean's model's focus on aspects that were surprising or shocking make it different from most models and presents the opportunity to carve out future plans and work from the final reflective stage. It is important to note that the critical reflection is for the use of the method itself, rather than how successful the process was in aiding the analysis. For the purpose of demonstrating this process, critical reflections will take place after reviewing the results of the method for both case studies, allowing us to draw comparisons between the two examples offered. When reflecting, we were particularly interested in the speed, clarity and accuracy of the processes and outputs.

## RESULTS

Herein, we present the use and reflection of the computational methods, by example of our two chosen case studies, using the aforementioned process. This is organised by the method.

### Topic modelling
### Set expectations (Step 1)
#### NHS COVID-19 app

One of the expectations of using these methods for this case study was to see the broad themes associated with the NHS COVID-19 app that were being discussed online. It was anticipated that each topic would be generated with a distinct set of words that would be associated with it to make it clearly defined. We also expected that the lexical items found within each theme would make it easy to label the topics.

#### Ofqual algorithm

For this case study, the expectation of applying these methods was to, once again, see whether there were any broad themes within this online discourse relating to the Ofqual algorithm. We anticipated a smaller number of latent topics in comparison to the COVID-19 app case study due to the reduced time frame and dataset size. We again believed that the lexical items would clearly indicate the overarching topic labels.

### View as trajectories (Step 2)
#### NHS COVID-19 app

Three latent topics were discovered through gensim LDA. Each topic contained ten key lexical items. These words are presented in descending order of association with the latent topic in Table 1. The number of topics was decided on through manual topic inspection and regeneration, examining the ten key words each time, to ensure minimal lexical item overlap.

Data should be presented as a trajectory. With regard to how the topics presented themselves in the tweets from each month of the research time frame, Fig. 1 details the percentage of tweets relating to each topic per month.

The discovery of these topics through the use of gensim's LDA function provides starting points for further focus. As shown, the most featured word of the most prominent topic is 'isolate'. This foregrounds the importance of the topic of self-isolation as part of the discourse surrounding this specific contact-tracing app, underlining that this disruptive effect the app can have on people's lives is dominating the social media discourse about it too. It is also of note that 'serco' is the most common word associated with topic two, which could show concern for who is responsible for the design and implementation of the app.

With the plotting of each topic's prevalence in the discourse for each month, it can be seen that Topic 1 is the topic that has been detected in tweets most consistently, although tweets were more concerned with Topic 2 at the time of the app's launch. In November 2020, there appears to be a rise in tweets that discuss Topic 3, and this is the same again in April 2021.

**Table 1**  **Ranking of the top 10 lexical items associated with each latent topic.**

|    | Topic 1 | Topic 2 | Topic 3 |
|----|---------|---------|---------|
| 1  | isolate | serco | download |
| 2  | positive | government | protect |
| 3  | contact | phone | help |
| 4  | work | work | store |
| 5  | notification | data | risk |
| 6  | code | download | google |
| 7  | phone | iphone | apple |
| 8  | results | private | play |
| 9  | self | phones | love |
| 10 | tested | good | available |

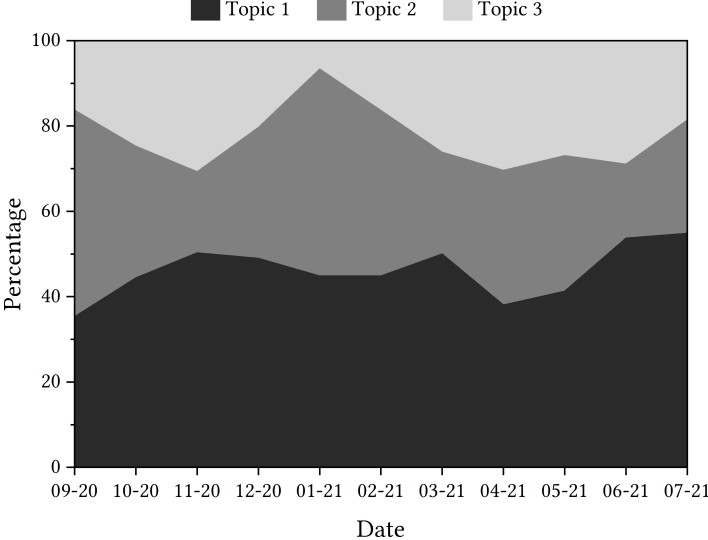

**Figure 1**  **Trajectories of topics detected in tweets containing 'NHSCOVID19app'.**

### Ofqual algorithm

Once again using gensim LDA, four latent topics were discovered through gensim LDA, each with ten key lexical items that are associated with that topic, presented in Table 2. Once again, the number of topics was decided on through manual topic inspection and regeneration. Initially, we had three topics, but we concluded a fourth was necessary due one topic containing many generic words and dominating the discourse.

Topic trajectories are presented in Fig. 2.

The one of the most featured words of the most prominent topic is 'government', which may foreground the importance of the role of the UK government in the decision to use and then withdraw the algorithm. 'Flaw' is the most featured word in Topic 2, potentially highlighting there is a significant amount of discussion about whether the algorithm was fit for purpose.

**Table 2 Ranking of the top 10 lexical items associated with each latent topic.**

|  | Topic 1 | Topic 2 | Topic 3 | Topic 4 |
|---|---|---|---|---|
| 1 | level | flaws | exam | students |
| 2 | government | statistics | gcses | levels |
| 3 | data | father | fiasco | schools |
| 4 | algorithms | foresaw | mutants | teachers |
| 5 | williamson | punishment | boris | grade |
| 6 | like | exams | johnson | school |
| 7 | wrong | labour | news | government |
| 8 | education | unlawful | blames | downgraded |
| 9 | blame | controversial | bbc | based |
| 10 | gavin | williamson | ofqual's | teacher |

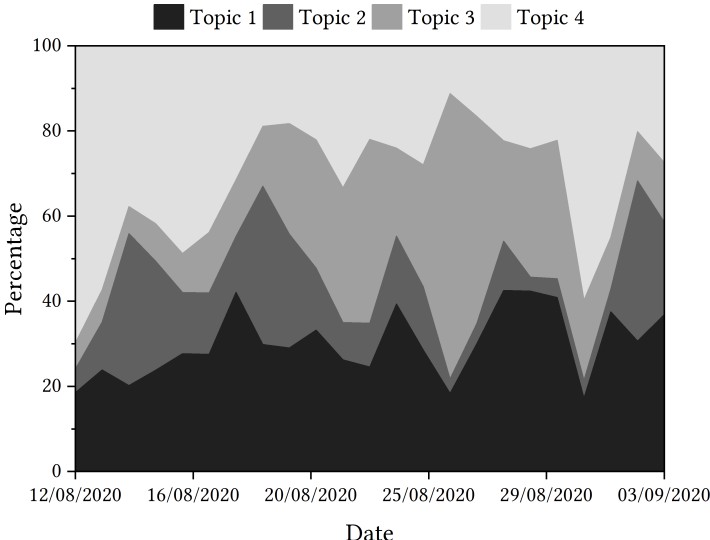

**Figure 2 Trajectories of topics detected in tweets relating to the Ofqual A Level algorithm.**

When looking at how the trajectories of the topics fared over the sample period, Topic 4, with words such as 'students', 'schools', 'teachers', etc., dominated the discourse initially. This aligns with the first reporting of the story, where lexical items such as these may have been popular. It might also point to intense focus on the feelings of students, teachers and schools who were affected by the algorithm. The topic did not dominate the discourse again until the end of the month, although, when examining the contextual factors of the algorithm in mainstream media, there is no clear reason why this may have happened.

### Human review (Step 3)
#### NHS COVID-19 app

After this, two blind human reviews were completed. A random sample of 10 tweets per month (110 total) were selected and categorised according to the pre-defined topics that were generated. The reviews found a 57% match between the human reviews and the automated topic labelling. Inter-annotator agreement (measured by Cohen's Kappa) was 0.525, indicating moderate agreement according to *Viera, Garrett et al. (2005)*. This agreement rate may have been due to the broad topics. In this, common errors included labelling of Topic 1 when the automated labelling suggested it would be Topic 2 (and vice-versa).

#### Ofqual algorithm

For the Ofqual Algorithm dataset, 10 tweets were randomly sampled for each day of the discourse (230 tweets total). The human reviews matched the automatically assigned topic 36% of the time. Inter-annotator agreement was 0.476, again indicating moderate agreement. A reason behind this might have been the separation between two topics that occurred in early stages of the process. The most common error seen here was the human reviewer labelling a tweet associated with Topic 1 as Topic 4 (and vice-versa). This low classification accuracy when comparing the human reviews to the automated topic might indicate that some of the topics are unclear or not fit for purpose.

### Examine items of interest with context (Step 4)
#### NHS COVID-19 app

The next step was to zoom in on items of interest with context. The sharpest monthly increase in a topic's discussion is the rise in tweets discussing Topic 2 in January 2021. When attempting to relate this contextually to the state of the pandemic in the UK, the shift to a more prominent Topic 2 came at the same time as the second nationwide lockdown and, thus, tweets that discuss the government may have increased. This was evidenced when looking at the human review sampled tweets. As shown in Fig. 1, the greater discussion of Topic 2 came at the expense of Topic 3, which was less prominent in tweets in January 2021. This suggests that tweets were less concerned with downloading the app and more concerned with the development of the app by the government and Serco at this point in time. Again, this was seen in the human review sample.

#### Ofqual algorithm

Discussion around the government and officials within the Department for Education, including Gavin Williamson, as seen in Topic 1, proves popular throughout the sample period according to the data. However, this is very different to how Topic 2, including lexical items such as 'flaw' and 'statistics', is discussed. This topic is popular at the start of the sample period, but declines in proportional popularity until the Educational Select Committee at the beginning of September. Instead, the discourse shifts to the discussion of Topic 3, featuring word such as 'fiasco' and 'mutant'. This topic is seen to gain in popularity after the government u-turn on the use of the algorithm, and is very prevalent in tweets up until the end of August. Upon the announcement of Boris Johnson labelling

the algorithm as 'mutant', this became the overwhelmingly most discussed topic according to the data. The labelling of the event as a 'fiasco' as the story gained in media popularity may have been a contributing factor in the rise of this latent topic, which was reflected in the language seen in the human review sample tweets too.

## Critical reflection (Step 5)

**Sunshine**   Upon reflection, the positive aspects of using LDA and topic modelling for both datasets have been the ability to see which lexical items appear frequently with one another, in an attempt to discover latent topics. The gensim tool for topic modelling was also easy to use and compatible with both datasets. Additionally, combining this method with the context of the chosen case studies illuminated topics for potential further analysis and follow-up.

**Rain**   One of the main things that did not work as well with using gensim's LDA topic modelling was that there was limited specific guidelines on how the output (*e.g.*, the lexical items) should be interpreted. As a result, our own interpretation has led us to consider what the topics might be about (although we have refrained from naming the topics in either case study). This means that comparing our findings to those conducting similar studies may be more difficult.

**Lightning**   A surprise that occurred through using LDA was the recurring lexicon that appeared in different topics in both datasets. This foregrounds the importance of context in this process: the same word can have different semantic and pragmatic meanings attached to it when it appears in two different topics (*e.g.*, 'work' as in 'function' or 'paid labour').

**Fog**   An aspect that caused slight confusion through using gensim's LDA is, once again, the interpretation strand. Although interpretation is encouraged after compiling a lexicon associated with topics, it is difficult to pin down how interpretation occurs. Given that LDA is an automated process based on frequencies, a challenge is how a human interprets the results to create meaning.

## Sentiment analysis
## Set expectations (Step 1)
### NHS COVID-19 app

The expectation of using sentiment analysis was to gain an overview of the discourse and see whether trajectories aligned themselves with contextual factors occurring simultaneously. The sentiment analysis results should tell us whether the overall feeling of this discourse is positive, negative or neutral.

### Ofqual algorithm

We again wanted to see the general feeling of sentiment detected in the discourse and how co-occurring events may have impacted sentiment. This sentiment may manifest in whether individual parts of the discourse are predicted to be positive, negative or neutral.

## View as trajectories (Step 2)
### NHS COVID-19 app

From the TextBlob sentiment analysis, Fig. 3 shows that overall sentiment was 0.03 to 0.16, indicating that overall sentiment is slightly above neutral.

After applying TextBlob sentiment analysis, data was presented as a chronological trajectory. The general trend saw positivity detected within tweets rise from September to November 2020, only for tweets to be categorised as more negative in both December 2020 and January 2021. Positive sentiment detected rose again in February and March 2020, dipping slightly in April, but rising again in May and June. Tweets were deemed less positive comparatively in July.

### Ofqual algorithm

From the TextBlob sentiment analysis, Fig. 4 shows that overall sentiment ranged from 0.088 to −0.052, indicating that overall sentiment is neutral. However, from the VADER sentiment analysis, Fig. 4 shows that overall sentiment ranged from 0.03 to −0.5, indicating that overall sentiment is negative.

When presenting the results as a trajectory, the general trend saw an increase in negativity on 14th August, with a steady rise in positivity detected in tweets for the next few days. From 17th August, there was an increasing in negativity detected within the tweets until 23rd August, when tweets were more positive. There was a rise in more negative tweets detected in late August but positive sentiment rose again towards the end of the data collection period.

## Human review (Step 3)
### NHS COVID-19 app

For this human review, 10 tweets per month (110 total) were randomly sampled and classified by two reviewers according to whether they were positive, negative or neutral. The human review score matched the computer assigned sentiment category on 50% of occasions. In terms of individual categories, the reviewers mostly disagreed with the tweets classified as 'positive', with only 22/54 occasions matching the algorithm-generated category. The inter-annotator agreement was 0.62, indicating substantial agreement. Between the reviewers, classifying tweets that the algorithm deemed 'positive' caused the most disagreement, with the reviewers not matching on 16/54 occasions. However, neutral tweets also caused disagreement, with 5/18 classifications not matching between reviewers.

Between the two reviewers, 33.6% of tweets contained negation structures. According to the human reviewers, 56.7% of these tweets containing negation structures were classified incorrectly by TextBlob. This will be explored further in Step 4.

When considering sarcasm, 9% of the tweets reviewed were labelled as sarcastic. 90% of these tweets were classified incorrectly as positive by TextBlob according to the human review. The qualitative examination of these tweets will be expanded on in Step 4.

### Ofqual algorithm

Again, 10 tweets per day (230 total) were randomly sampled and classified by two reviewers according to whether they were positive, negative or neutral. The human review score

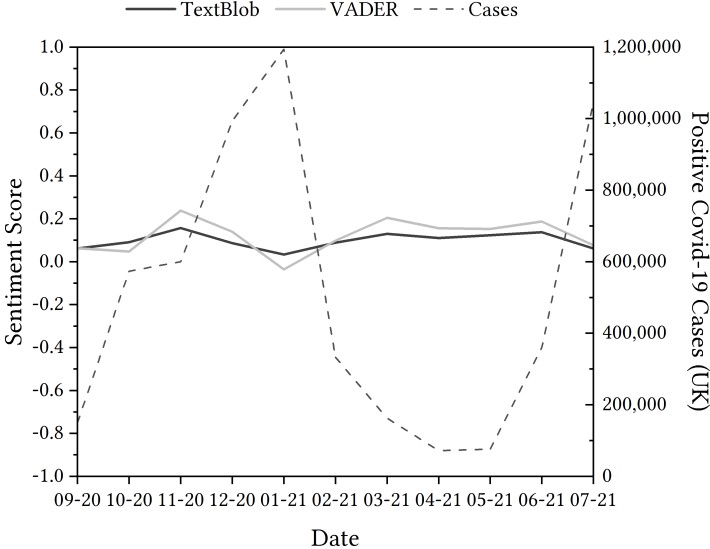

**Figure 3** Evolution of the sentiment of tweets containing 'NHSCOVID19app' using TextBlob and VADER from September 2020 to July 2021.

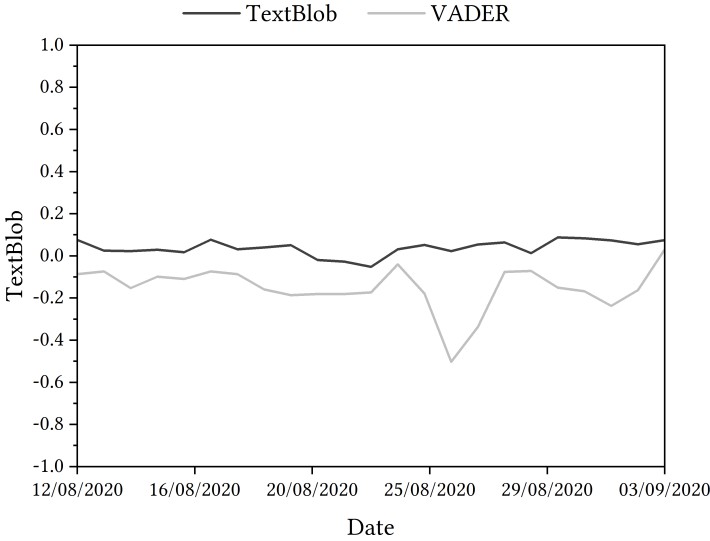

**Figure 4** TextBlob and VADER sentiment analysis of tweets relating to Ofqual A Level algorithm in August and September 2020.

matched the computer assigned sentiment category on 40% of occasions. The inter-annotator agreement was 0.547, again indicating moderate agreement. This agreement may have been slightly higher than the topic modelling agreement levels due to the more straightforward classifying process (positive, neutral, negative only). Nevertheless, there were still disagreements between reviewers. Between them, classifying tweets that the

algorithm deemed 'neutral' caused the most disagreement, with the reviewers not matching on 29/72 occasions.

A total of 26% of tweets were found to contain negation structures. According to the human reviewers, 74.6% of these tweets containing negation structures were classified incorrectly by TextBlob. Considering sarcasm, 6% of the tweets reviewed were labelled as sarcastic. 71.4% of these tweets were classified incorrectly as positive by TextBlob according to the human review. A closer look at these tweets takes place in Step 4.

## Examine items of interest with context (Step 4)
### NHS COVID-19 app
Comparing sentiment detected in tweets relating to the app to the wider context of the UK's history with the pandemic was the next step in the process. For this, we undertook an examination of positive COVID-19 cases and the sentiment detected in tweets, depicted in Fig. 3 as they showed similar inverted trends—sentiment for the app is more negative as the number of cases rise in January, and sentiment for the app is more positive as the cases lower in the spring months. Note that we don't want to suggest causation here, but to highlight the importance of wider contextual data which are vital to interpret sentiment analysis results at the time at which the data was recorded.

Additionally, from the human review sample selected, there were a number of tweets which had been classed as positive (1.0). Of these tweets, there were some of note that had this classification but, upon inspection, it is likely that other human classifiers would agree with us in making an alternative decision, and decide that they were expressing mainly negative sentiment instead. An aspect of language that these tweets have in common is the inclusion of negation within their syntactical structures (*e.g.*, "I don't"; "can't say"; "not to be")', together with words that taken without negation could be interpreted by an algorithm as very positive (*e.g.*, "trust", "best", "impressed", "proud"). Tweets that could be interpreted as sarcastic were also of note here for similar seemingly inaccurate detection reasons. Several tweets in the data sets were all categorised as tweets that had strong positive sentiment detected when classified using the TextBlob module but were categorised as negative when examining with context.

The seemingly incorrect categorisation of these tweets could mean that the reliability of the TextBlob sentiment analysis tool may be questioned. Therefore, we deployed another sentiment analysis module for further investigation. Due to the fact that the VADER sentiment analysis module states that it accounts for negation, it was a logical next step to see whether there was a difference in the categorisation of the dataset, but also with these focused tweets more specifically.

When comparing the two, as shown in Fig. 3 the general trend of the data when comparing the VADER sentiment analysis to the TextBlob sentiment analysis is similar, but varies in two different places: October 2020 and May 2021. There are opportunities to investigate these differences in sentiment polarity detection through a critical lens, but it could be said that sentiment analysis alone might offer limited capability to do this without the inclusion of a qualitative method in conjunction. That being said, the comparison

could still be argued as noteworthy: the polarity of VADER appears to be more extreme than TextBlob—seeing deeper rises and falls in sentiment detected in tweets.

With regard to the tweets that were categorised as positive by TextBlob, despite finding negation in the tweets, these were classified as −0.4023, 0.6369, 0.4767 and 0.2924. This suggests that, while there is some improvement in detecting negation within VADER, both VADER and TextBlob and other easily accessible sentiment analysis models may still benefit from further improvements of this aspect of language to improve the accuracy of their tools; but perhaps more importantly, users of these methods like us would benefit from greater transparency regarding these limitations of the models.

For tweets that may be sarcastic, the sentiment detected was still positive with the VADER module, with values of four particular tweets being 0.8316, 0.7622, 0.6767 and 0.6958 (rather than 1.0). Again, this suggests that, even though the VADER system was potentially able to detect sarcasm better than TextBlob, both systems may benefit from further focus on this. Additionally, users would benefit from more transparency regarding these important limitations.

### Ofqual algorithm

To interpret the results as best as possible, they were examined with key dates in the chronology of the Ofqual algorithm, as per the next step in the process. For example, a negative change in sentiment was detected on August 14th, the day after the results were shared with students. A rise in positive sentiment detected on 17th August came on the same day as the government announced that the algorithm-calculated grades would be replaced with teacher assessed grades. The sharp negative change in sentiment on 26th August came on the same day as UK Prime Minister Boris Johnson told students that their results had been affected by a 'mutant algorithm'. A rise in positive sentiment on 3rd September came the day after Ofqual Chair, Roger Taylor, apologised to students when appearing at the Educational Select Committee at the House of Commons.

Similarly to the COVID-19 app case study, there were instances where the algorithm classification involving negation and sarcasm could be seen as incorrect. Within the dataset, tweets that were categorised as positive did include negation, pairing 'not' with 'believe' and 'no' with 'accident', for example. Once again, there were many tweets that could be deemed to be sarcastic that were classed as positive by TextBlob. With these tweets, hyperbolic adjectives such as 'sophisticated', 'great' and 'flawless' all seen.

As a result, the VADER module was deployed. With the inclusion of the VADER sentiment data, as displayed in 4, overall sentiment ranged from 0.03 to −0.5, indicating that sentiment is negative. On 23rd August, there was an abrupt change in sentiment detected in tweets, as the data in Fig. 4 shows—sentiment detected increased in positivity. This is perhaps more pronounced when looking at the shift in sentiment that follows in the coming days—a change in sentiment to −0.5 on 26th August. Tweets were deemed more positive comparatively at the beginning of September.

When revisiting the tweets in the sample that included negation but were potentially incorrectly classified, new VADER scores were 0.8372, 0.6546, −0.1655 and 0.4019, showing a similar result to the COVID-19 app's VADER scores for negation specifically. For tweets

deemed sarcastic, the VADER scores were 0.9739, 0.3612, 0.1548 and 0.9081. Once again, this replicates the findings from the COVID-19 app case study.

## Critical reflection (Step 5)

**Sunshine** A positive aspect of using this method is the ability to analyse a large amount of data within a seemingly short amount of time, with a straightforward coding process. This was seen in both case studies. Another aspect that worked well was how simple the two methods were to compare after deciding to integrate VADER into the sentiment analysis process. Sentiment scores over a larger period time, such as shown in Figs. 3 and 4 provide an instant overview of the trend of sentiment (*i.e.,* 'going up and down') in the dataset over time. This lends itself to identify 'turning points' for further investigation, potentially through in-depth qualitative methods.

**Rain** An aspect of investigation that did not work as well was the fact that negation and sarcasm are still unsolved challenges in sentiment analysis and may therefore decrease the accuracy of the techniques and the robustness of the findings. Individual 'sentiment scores' are hard to interpret and feel meaningless on their own.

**Lightning** A surprising occurrence when undertaking sentiment analysis using TextBlob and VADER was the variation between outcomes seen, as it poses questions as to which is the more reliable tool to use for conducting a sentiment analysis at this scale. Additionally, for the Ofqual algorithm, the variation seen in the data—where sentiment changes dramatically—was surprising, particularly when contextually there was little that could be illuminated with some of the changes (like 24th August).

**Fog** A confusing part of using sentiment analysis comes from the idea of interpreting the data. When a sentiment score is generated, there is limited guidance on what that score means and how it can illuminate further insights into the status of the contact-tracing discourse in the UK. Although these scores can be used as a starting point, presenting them without context may leave a lack of clarity. For the Ofqual algorithm, there is limited guidance on what a sentiment score means and how it can illuminate further insights into the status of the discourse, leading to a lack of clarity. For example, what is the difference between a score of $-0.18091$ and $-0.1815$, as seen on 21st and 22nd August in the VADER classification for the Ofqual algorithm?

## Emotion detection
## Set expectations (Step 1)
### NHS COVID-19 app

The expectation of using emotion detection was that using this method would allow an insight into how people felt about the app and whether there were any shared or common emotions expressed. We also expected the results to tell us which emotions were more prevalent at certain times within the longitudinal discourse.

### Ofqual algorithm

Our expectation was to see which emotions were the most common in the discourse and how the emotions detected may change over time. With this being a shorter time period,

we expected to see perhaps not as many emotions detected or possibly not as much gradual change in emotions predicted.

### View as trajectories (Step 2)
#### NHS COVID-19 app

For the next step in the process, the data presented in the trajectory displayed in Fig. 5 shows that 'trust' was the emotion most detected in tweets relating to the app, followed by 'fear' and then 'anticipation'. By examining the percentage of tweets that detected each emotion, the rise in tweets related to 'fear' at the end of 2020 could be deemed to be of interest, as could the rise in tweets related to 'anticipation' in the spring of 2021.

#### Ofqual algorithm

Once again, as shown in Fig. 6, 'trust' was the emotion most often detected, followed by 'fear'. Looking at the trajectory of emotions detected in the sample period, there are key dates that appear to have greater spikes in 'fear', and thus a downturn in 'trust' detected, including the days following the u-turn announcement, the statement about 'mutant algorithm' and 30th August. However, once again, it remains unclear whether tweets relating to 'trust' are indeed positive indicators of trust or indicators of mistrust.

### Human review (Step 3)
#### NHS COVID-19 app

For the COVID-19 app, ten tweets per month (110 total) were randomly sampled to be reviewed. The categories to be assigned were 'trust', 'fear', 'anticipation', 'anger', 'surprise', 'sadness', 'disgust', 'joy' and 'no emotion'. Reviewers matched the EmoLex assigned category on 25% of occasions. The inter-rater reliability was 0.44, indicating moderate agreement. These agreement levels may be lower than the topic and sentiment agreement levels due to the range of emotions available to classify with. Within this, between the reviewers, classifying tweets that the algorithm deemed as 'anger' caused the most disagreement, with the reviewers not matching on 5/11 occasions. Reviewers categorised these tweets as 'fear' or 'disgust' instead.

#### Ofqual algorithm

For the Ofqual discourse, ten tweets per day (230 total) were randomly sampled to be reviewed. Once again, the categories to be assigned were 'trust', 'fear', 'anticipation', 'anger', 'surprise', 'sadness', 'disgust', 'joy' and 'no emotion'. Reviewers matched the EmoLex assigned category on 37% of occasions. The inter-rater reliability was 0.481, again indicating moderate agreement. Once again, these agreement levels may be explained by the range of emotion options available. Tweets reviewed in this process will form the examples of the following section. Between the reviewers, classifying tweets that the algorithm deemed as 'fear' caused the most disagreement, with the reviewers not matching on 17/60 occasions. Instead, reviewers categorised these tweets as 'anger', which is the reversal of what was seen in the COVID-19 app dataset.

### Examine items of interest with context (Step 4)

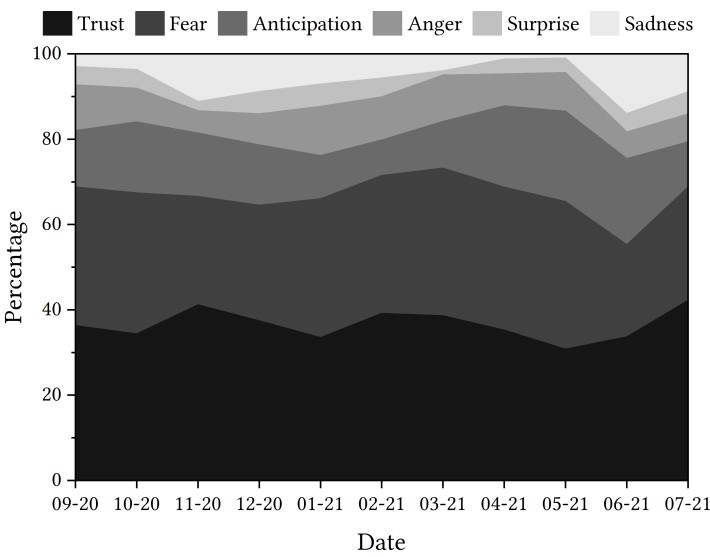

**Figure 5** **Emotions detected in tweets relating to 'NHSCOVID19app'.**

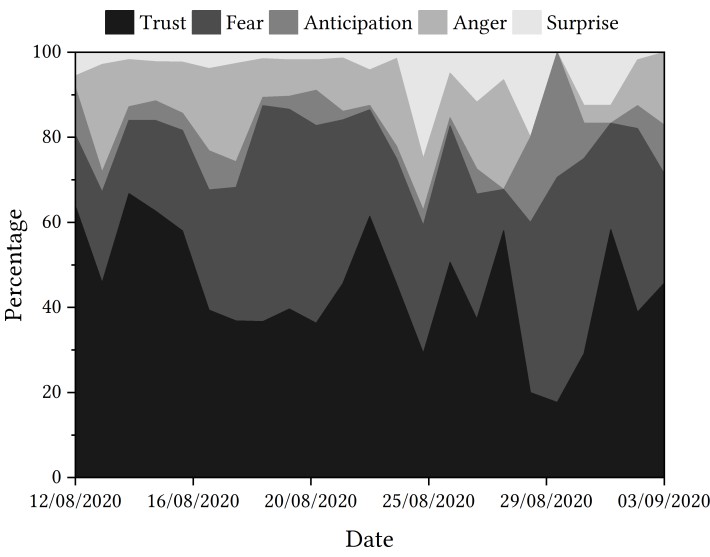

**Figure 6** **Emotions detected in tweets relating to the Ofqual A Level algorithm.**

### NHS COVID-19 app

When zooming in, the consistency of 'trust' being an emotion that is detected within tweets relating to the app means that it is a prominent emotion of discussion. With the premise that 'trust' and 'fear' are separate emotions, it might be assumed that the tweets relating to 'trust' are indicative of the person trusting the app, rather than not trusting it. This thought guided us through the next part of the process.

Some tweets had been classified as containing the emotion 'trust' within them; again, we would have categorised these differently due to the opposition to trust noted within each

tweet. The tweets included the word 'trust', which had been negated using the contracted modal verb 'wouldn't'. This may have not been detected by the EmoLex module and, instead, the word 'trust' superseded any other vocabulary as the classifier made its decision. This is similar to the negation issue using TextBlob for sentiment analysis. However, there are other tweets that have been noted as having 'trust' detected within them that do not contain the word 'trust'. Although it may be possible to argue that the words 'faith' and 'encourage' could hold some similarity in meaning with the word 'trust', there is no explicit mention of 'trust' within these tweets and a human might categorise them as expressing a lack of trust in the app, again as negations are present (*e.g.*, "not downloading", "haven't got faith"). This further demonstrates a possible potential limitation of using tools such as EmoLex for emotion detection.

### Ofqual algorithm

One potentially interesting emotion to explore is 'anticipation'. While there were some tweets throughout the discourse that were categorised as expressing the emotion 'anticipation', this is a relatively low amount of tweets until 30th August, when 11.63% of tweets were classified in this category, rising sharply from 4.17% the day before and no tweets detected two days before. Once again, 30th August does not appear to be a significant date in the timeline of events relating to the Ofqual algorithm, and so the potential rationale for this increase is not immediately clear.

'Anger' and 'surprise' are the two other emotions detected in the dataset. 'Anger' appears to be more prevalent in the discourse at the start of the sample period, particularly on the day that results were released and the day of the government u-turn. With 'surprise', however, this emotion is detected in relatively few tweets at the start of the discourse, which seems to change on 26th August—the date that Johnson made a statement on the 'mutant algorithm'. This 'surprise' might be attributed to the use of the word 'mutant', or the general unexpected nature of the statement. In what might seem an apparent consequence of this statement, 'surprise' is generally detected more frequently that previously in the majority of the remaining days in the sample period.

As with the COVID-19 app case study, there are questions as to whether the emotion 'trust' indicates the direction of trust. For example, tweets containing 'criticised', 'unfair', 'unequal' and 'failure' were all attributed to 'trust', despite being potentially more indicative of distrust instead. Additionally, when presented with a tweet that contained vocabulary such as 'great' and 'similar' (used in a literal and not sarcastic context), there was limited opportunity to classify this as 'happy' or 'supportive'. For context, EmoLex had classified this tweet as 'anticipation'. Going forward, this might be a consideration for developing expectations when working with emotion detection algorithms, which feeds into the process.

## Critical reflection (Step 5)

**Sunshine**  One of the aspects that worked well here was the speed of automated detection with a large dataset and that each tweet was able to be classified in some way.

**Rain**  As with sentiment analysis, the accuracy of the detection may have been an issue again when deploying the EmoLex emotion detection module. Additionally, insights into

the discourse are presented without context—something that would have been helpful in the analytical process. With limited context, the emotions could be seen as arbitrary.

**Lightning**  A surprising element of using emotion detection was the inclusion of 'positive' and 'negative' within the initial emotion set in EmoLex. This might have meant that vital information may have been missed as the tweets may have been closely aligned with other emotional categories, yet this has not been included in the results.

**Fog**  Something that could have been clearer when using this tool was the distinction between emotions. For example, should tweets associated with trust only feature those that actively support this emotion, rather than oppose it? As seen in further exploration, there were tweets categorised as containing the emotion 'trust' that may have been categorised to be in opposition to trusting the app if classified by a human. If this had been clearer before using the tool, the picture of the discourse may have been more accurate.

## DISCUSSION

The following section will take the form of two main parts. The first part discusses each step of the process. The second part looks at the implications that our research has on current and future research in the field of social media research.

### The analysis process
#### Set expectations (Step 1)
Through the setting of expectations, we were able to make clear what we aimed to discover through using the methods on each case study. These were particularly useful to refer to when undertaking the critical reflection as a measure of what we anticipated might be unearthed during the analysis of results, which supports the advocacy of *Post, Visser & Buis (2017)* on this topic. This will be revisited in the discussion of the critical reflection later.

#### View as trajectories (Step 2)
The analysis and trajectories, presented by the method used, will now be discussed in detail. Displaying the data as chronological trajectories was helpful in identifying changes in the discourse with regard to differing topics and emotions, and how sentiment fluctuated. Even though *Howard (2021)* advocated that this is useful for longitudinal datasets and discourses, this was seen as beneficial to both the longitudinal COVID-19 app discourse and the snapshot A Level algorithm discourse.

   Topic model analysis, as a whole, used adjacent words well to describe latent topics present within the text, which is similar to how *Hu, Chancellor & De Choudhury (2019)* used LDA to aid their investigation into homelessness discourses online too. Additionally, gensim's ease of use and compatibility with the Twitter dataset meant that it was a good fit to begin the analytical process with this particular social media platform, much like *Hidayatullah et al. (2019)* and their use of gensim. Seeing the topics presented as trajectories allowed us to use them as a springboard for further investigation, like *Song et al. (2019)*. However, the process for deciding how many topics were chosen for each discourse could have been improved and may have impacted some of the errors found later on with classification. For example, in the Ofqual discourse, we manually decided to expand from

three topics to four (*Nguyen et al., 2020*), but instead we could have used a measure for topic coherence, such as Hierarchical Dirichlet Process (*Teh et al., 2004*), to arrive at an optimal number.

For sentiment analysis, the trajectories showed a large amount of data that was able to be analysed through both TextBlob and VADER, which was the starting point in gaining some insights. The comparison between the two systems, as another was required, was easy to facilitate too, much like how *Pak & Paroubek (2010)* compared classifiers. However, these trajectories could only tell us a small amount of information to begin with, and it was clear that more interpretation and analysis was necessary. Nevertheless, it was good to know where to focus our attention moving into the next steps (where items of interest are examined with context).

Reflections regarding emotion detection are similar to those concerned with sentiment analysis. The ease of deployment and the fact that categorisation could occur for every tweet was an advantage of using this method, which meant that displaying this as a trajectory aided our identification of items of interest. This aided the showcase frequently detected emotions, which links to earlier strengths of the method discussed in the work of *Mathur, Kubde & Vaidya (2020)* and *Aribowo & Khomsah (2021)*.

### Human review (Step 3)

Undertaking the human review meant that the algorithm-generated results could be compared against human classification. The results found a low percentage of classification accuracy when comparing the human and algorithm-generated scores, ranging from 25% to 50% match. This, at surface level, indicates that there are potential errors in the classification, which may be from either the human or algorithm score.

For topic modelling, the main finding was that the topics may not have been accurate, particularly for the Ofqual discourse, due to the low percentage accuracy (57% for the NHS discourse and 35% for the Ofqual discourse). As we followed the principles of *Nguyen et al. (2020)*, we thought that having fewer topics would have led to fewer inaccuracies. However, it may have been more beneficial to have refined the topics to create smaller, more-defined themes, like *Sengupta (2019)*. This was particularly true for the Ofqual discourse.

For sentiment analysis, this process not only allowed us to see the accuracy of sentiment detection, but also allowed us to see the proportion of sarcastic and negated tweets that were classified seemingly incorrectly. The figures of 56.7% (NHS) and 74.6% (Ofqual) of negated tweets and 90% (NHS) and 71.4% (Ofqual) of sarcastic tweets being classified seemingly incorrectly by the automated classification indicate that these issues were prevalent, much like the work drawn upon by *Gupta & Joshi (2021)* and *González-Ibánez, Muresan & Wacholder (2011)*. This was a similar process for emotion detection, instead highlighting the potentially incorrect classification of 'trust'. Both of these will be expanded upon in Steps 4 and 5.

The potential questionable output from the 'off-the-shelf' tools had implications for our analytical process. Agreement between human and algorithm classifications was better for the NHS App on two occasions (57% *vs* 36% for topic modelling, 50% *vs* 40% for sentiment analysis) and better for the Ofqual algorithm on one occasion (37% *vs* 25%

for emotion detection). Regarding subjectivity, the discrepancies may question what is understood to the 'ground truth' in these scenarios. More specifically, occasions where the human reviewers and the algorithm-generated score or classification were in disagreement could have been due to two reasons: ambiguity in the labels or the idea that some text holds meaning on multiple levels (*Fujioka et al., 2019*). These concepts are usually difficult for computational linguistics to detect and interpret. This meant that we proceeded with Step 4 with certain groups of tweets as immediate focal points, especially when concerned with changes to potential trajectories or the expectations previously set. When reflecting, this meant that the latter steps in the process may have been influenced by these low agreement rates.

The human review aspect could benefit from further development. As reported, there were occasions where the human reviewers did not agree on classification, with inter-annotator agreements usually being moderate. This was particularly prevalent when classifying neutral tweets in the sentiment analysis and classifying tweets relating to anger and fear in the emotion detection. Seeing that humans may disagree to a significant extent about posts belonging to the same classification, or a post's polarity, has implications for how annotation works in text mining more broadly. For instance, emotion categories that may overlap, such as fear and anger, may cause discrepancies (*De Silva et al., 2018*; *Leung et al., 2021*). Also, some reviewers classify tweets including 'positive' and 'negative' elements as 'neutral', whereas others may find the best fit, exemplifying the complexities of agreeing on sentiment (*Villena-Román & García-Morera, 2013*). Supporting reviewers with examples of lexicon, or more complex emotional categories (*Jiang, Brubaker & Fiesler, 2017*) that could be included in certain categories through training may mitigate such errors and encourage objective discussion.

Additionally, although we were able to see where things may have been classified potentially incorrectly, a formal approach was not undertaken here. Fortunately, there are studies within social media studies that use computational linguistic methods that have included a human review aspect, whether formalised or in a more informal approach, such as *van Atteveldt, van der Velden & Boukes (2021)* and their use of human-trained coding that outperformed machine learning approaches. The inclusion of a more formal qualitative aspect here would aid the analysis and categorisation in the human review stage. As a result, a formal qualitative method could be used, such as critical discourse analysis, which would aid classification through appropriate supporting theoretical frameworks.

### Examine items of interest with context (Step 4)

Using contextual information to illuminate the analysis of the discourses proved rewarding and challenging. Using contextual information, at the suggestion of *Agarwal et al. (2015)*, meant that more meaning was created from the results. This was particularly pertinent in the Ofqual algorithm case study due to the more focused timeline. Zooming in closely on these items of interest that were identified in the Step 3 also provided further insight into the potential accuracy of the methods used. For topic modelling in this stage, the limitations and challenges recognised were collectively aligned with how to interpret the

results to produce more meaningful insights. This may be seen to build on the previous work by *Maier et al. (2018)*.

For sentiment analysis, with the limited guidance on what the scores reflect or mean, this adds little to the earlier findings of the study by *Pokharel (2020)* on the Twitter response to COVID-19 in India due to the limited distinction in results when using sentiment analysis alone. Therefore, our findings here are more in agreement with *Sivalakshmi et al. (2021)*, with their assertion that TextBlob being unable to read tokenized special characters may have hindered its performance and suitability here, and *Chauhan, Bansal & Goel (2018)*, who believe VADER's sensitivity to social media formats make it the more appropriate choice. Additionally, with the limited guidance on what the scores reflect or mean, this adds to the use case of sentiment analysis. This reinforces the points made in studies such as *Jiang, Brubaker & Fiesler (2017)*. The fact that this was only illuminated through our critical reflection process shows that there is more work to be done in understanding the limits of sentiment analysis, echoing the work of *Stine (2019)*.

When examining with context in the emotion detection output trajectories, it raised issues as to whether whether trust and distrust should exist within the same emotional 'trust' umbrella in EmoLex, echoing the need for more distinct emotional categories (*Jiang, Brubaker & Fiesler, 2017*). The reflections on EmoLex in this contribution are, therefore, more in line with the improvements suggested by *Balakrishnan et al. (2019)*, *Balakrishnan & Kaur (2019)* and *Fast, Chen & Bernstein (2016)* that concern the intricacies of emotions.

### Critical reflection (Step 5)

The use of Maclean's critical reflection weather model provided a format for the process of discovering advantages and limitations of the methods by providing specific focal points. The advantages it highlighted were that the examined computational methods were easy to implement and provided a starting point for further investigation into the discourse. Exploring how views expressed altered through the course of the app's existence, and viewing these as linear trajectories, aided the investigation. The limitations of the methods that the reflection highlighted include the instances of diverging interpretations when examining linguistic features in context; this means that the accuracy of the methods may be brought into question. This was the case for sarcasm and negation in sentiment analysis, the naming and accuracy of topics in LDA (introducing biases as mentioned by *Saura, Ribeiro-Soriano & Saldaña, 2022*) and the distinction between emotions in emotion detection. It is possible that a different model may have been able to mitigate these potential inaccuracies, such as how VADER outperformed TextBlob in the sentiment detection of negation; however, these models may not be as accessible to social media researchers.

## Implications for research

This study has several implications for social media researchers wishing to investigate the state of online discourses using descriptive and predictive computational linguistic methods. Firstly, the examination of these methods through a critically reflective lens echoes some of the existing social media analysis critiques of topic modelling, sentiment analysis and emotion detection algorithms.

On a related note, the process we have formed, by combining other models and ideas for existing best practice, may be developed into something more formal and robust: it has the potential to act as a foundation for those unfamiliar with these methods but may wish to use them for exploring public discourses. Social media researchers using this approach may apply these computational linguistic methods to analyse large social media datasets with confidence. It means researchers can realistically manage expectations through the logging of what the researcher wishes to achieve so this can be revisited in the critical reflection aspect, as well as doing more than looking at the outputs of the predictive algorithms by displaying trajectories and examining with context.

When considering future work, this should investigate the ways in which computational linguistic methods may be used more effectively by combining them with qualitative research methods, such as corpus linguistics or discourse analysis. This may help to mitigate some of the limitations of computationally descriptive and predictive analytical approaches that were found during the process of critical reflection in this article. The strong emphasis on context, particularly in critical discourse analysis, means that how views are expressed and related to the events occurring at the time can play a more prevalent role in the analysis of public discourses.

## CONCLUSION

In applying popular computational linguistic methods to Twitter case studies, we demonstrate the potential use of a process that combines existing best practice and critical reflections on the use of the methods for social media researchers, like us, who are not experts in computational linguistics. Our findings show that, by following this analysis process, we were able to do more with the results generated by the algorithms through the examination of contextual factors and chronological trajectories post-human review. Our critical analysis and reflection shows that the classification of tweets as expressing a certain sentiment or emotion is not always accurate, with negation and sarcasm standing out as linguistic phenomena that algorithms struggle with. This was reinforced by the issues highlighted with both human-algorithm agreement and inter-annotator agreement. The threads of trust and fear in both datasets, which were detected through emotion modelling algorithms, showed that there is deeper exploration needed to ascertain the cause of these emotions and the degree to which they are felt; this is possibly where a more discursive sociolinguistic method may be helpful. Finally, the use of Maclean's reflective toolkit enabled concise evaluations of the performance and suitability of each of the methods and raised potential concerns with interpretation and lacking transparency regarding limitations such as as dealing with negation and sarcasm. Overall, the critical reflections found from following this process should begin to support the confident and realistic use of computational linguistic methods by social media researchers. NLP approaches should be applied with care and a critical reflection on their output. While the analytical process that includes current existing best practices is a good start, more formal qualitative interpretation of results could be necessary.

### Funding

All authors are supported by the UKRI Trustworthy Autonomous Systems Hub (UKRI Grant No. EP/V00784X/1). Dan Heaton is supported by the Horizon Centre for Doctoral Training at the University of Nottingham (UKRI Grant No. EP/S023305/1). The funders had no role in study design, data collection and analysis, decision to publish, or preparation of the manuscript.

### Grant Disclosures

The following grant information was disclosed by the authors:
UKRI Trustworthy Autonomous Systems Hub: EP/V00784X/1.
Horizon Centre for Doctoral Training at the University of Nottingham: EP/S023305/1.

### Competing Interests

The authors declare there are no competing interests.

### Author Contributions

- Dan Heaton conceived and designed the experiments, performed the experiments, analyzed the data, performed the computation work, prepared figures and/or tables, authored or reviewed drafts of the article, and approved the final draft.
- Jeremie Clos conceived and designed the experiments, analyzed the data, performed the computation work, authored or reviewed drafts of the article, and approved the final draft.
- Elena Nichele analyzed the data, authored or reviewed drafts of the article, and approved the final draft.
- Joel Fischer conceived and designed the experiments, analyzed the data, authored or reviewed drafts of the article, and approved the final draft.

### Data Availability

   Raw data and code are available in the Supplemental Files.

### Supplemental Information

Supplemental information for this article can be found online at http://dx.doi.org/10.7717/peerj-cs.1211#supplemental-information.

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
