# Peer review of "Critical reflections on three popular computational linguistic approaches to examine Twitter discourses"

_PeerJ Computer Science, doi:10.7717/peerj-cs.1211_

## Round 0.1 · original submission · Major Revisions

Reviewers clearly value the work undertaken in this paper, however they do also have some important concerns, not least that the title and the contributions as presented in the paper are somewhat overstated. This would need either fleshing out the contributions or toning down the claimed contributions.

While the reviewers are overall positive about the value of the work, they also suggest a set of revisions which would require substantial work prior to resubmission. Still, I believe that this is doable work as long as authors are willing to undertake this work, hence I recommend a decision of major revisions.

Reviewer 1 ·

Basic reporting

==Writing==
Overall, the text is clear with the use of professional English. However, the writing felt rushed, for example:

- Many places with additional space before full stop (‘.’), e.g. lines 118, 128, 131, 150.
- Incorrect formatting of references, e.g. lines 86, 90, 98, 717, 771. Also check the reference on line 246.
- Unfinished sentences, e.g line 93, 343 (‘to For’), 765
- Quotes (use ` instead of ' for opening quotes)
- 410: ‘that the there’
- 48: 'the of these'
- 58: capitalize 'related'
- 719: should be ‘large’

==Related work==
The authors discuss related work wrt the 3 methods they use (topic modeling, sentiment analysis, emotion detection). I would suggest changing the focus of the related work section. There is numerous work on Twitter data and the use of these methods. The individual studies applying such methods on Twitter data that are discussed in this section seem like a somewhat arbitrary selection. I would suggest condensing this part. In contrast, I was hoping to see more discussion of papers that have critiqued these methods and/or offered guidelines on how to use these methods. This relates directly to the goal of this article, and it’s also important to understand how this article differs from existing ones.

Some related papers that could be useful:
- “Comparing Apples to Apple: The Effects of Stemmers on Topic Models”, 2016 https://aclanthology.org/Q16-1021/
- “How We Do Things With Words: Analyzing Text as Social and Cultural Data”, 2020 https://www.frontiersin.org/articles/10.3389/frai.2020.00062/full
- Text as Data: The Promise and Pitfalls of Automatic Content Analysis Methods for Political Texts, 2013 https://web.stanford.edu/~jgrimmer/tad2.pdf
- Beyond Accuracy: Behavioral Testing of NLP Models with CheckList, 2020, https://aclanthology.org/2020.acl-main.442/ (systematic test cases, they looked at sentiment analysis)
- Machine Learning and Grounded Theory Method: Convergence, Divergence, and Combination, 2016 https://dl.acm.org/doi/abs/10.1145/2957276.2957280

Furthermore, the related work section doesn’t connect well with the popularity of deep learning methods in NLP. For example, the sentiment analysis section focuses on dictionary-based approaches. It’s fine to use such an approach for this study, however the authors should acknowledge other approaches (e.g. for sentiment analysis), and motivate why they choose a particular approach.

Some statements lack justification or seem outdated, e.g line 114 "A staple of most sentiment analysis methods is the Naive Bayes classifier"

I didn't understand the reasoning in line 101: how is BoW closely connected to the previous sentence on bigrams?

Experimental design

The authors experiment with 3 methods and 2 datasets (COVID19 and Ofqual). Although this breadth is great, the analyses remain somewhat shallow, e.g. plotting time series of sentiment and topic measurements, qualitative inspection of the topics based on some highly ranked words, and relating observed trends to knowledge about the events.
- The identified weaknesses/challenges (e.g. negation, sarcasm) are mostly based on some qualitative inspection (e.g. lines 530-539, 646-656). It would have been much stronger if they had validated these tools for their datasets by annotating a sample of their tweets (e.g. a few hundred of each datasets) and categorizing the errors that were made.
- Similarly, when comparing different methods, a quantitative comparison of time series would be more convincing than comparing them visually (e.g. lines 545-547).
- Some analyses were based on a very small number of tweets (e.g. 4 tweets, 552-562, 586-590), I’m not sure what we can take away from that.

Some smaller comments:
- Overall the data collection and pre-processing steps are clear. The authors even indicate in the text which functions (e.g. from gensim) they used. Some clarifications are still needed, e.g. how exactly was the text pre-processed for the topic modeling? The text mentioned bigrams and lemmatization. I’d particularly like to know more about the lemmatization, as this can be quite tricky to do with language on Twitter which contains a lot of language variation. Furthermore, did the authors apply any thresholding on e.g. (document) frequency regarding which words to include? It would be helpful to state the vocabulary size.
- It was unclear whether the four steps (starting in 365) are based on previous work, or whether the authors themselves came up with these steps. The authors also don’t discuss how general this procedure is, e.g. ‘view as trajectories’ only seems to apply to research questions/data that involve a temporal component.
- I ’d like the Maclean model that uses 4 stages (Sunshine/Rain/Lightning/Fog) to facilitate critical reflection. The text states that it was developed outside of social media research, I’d be interested in knowing for what type of problems it had been developed.
- I couldn’t find how many topics the final topic models used (e.g. line 411 says “there would probably be no more than five latent topics…”) and how this hyper parameter was set. If there are more than the 3 topics displayed in Table and Figure 1, the authors should explain how they selected the topics to be included in the time series analysis.
- I liked the observation that interpreting sentiment scores is difficult!
- I’d avoid any use of ‘significant’ unless some statistical test was used (e.g. line 632 ‘could be deemed significant’)

Validity of the findings

Code and data is provided.
Overall additional documentation would be useful. E.g. a README describing what can be found in each file, the meaning of the column names in the data files (even though many of them are self explanatory). For the code, it's useful to know which libraries (e.g. which versions) one needs to install.

Additional comments

- The introduction as well as the last page describes a link to explainable AI. It’s not clear to me how this paper relates to this line of work.
- I think the title is too general and overclaiming the contribution of the paper. The paper only looks at 3 different tasks and a small set of methods to perform these tasks. The methods themselves are available through easy to use tools but some of them are not state-of-the-art methods in NLP. I would suggest making the title more specific wrt the methods/datasets used in this paper.

·

Basic reporting

The paper has a clear but at times sloppy writing style, with at points sentences that stop in the middle to proceed with a new sentence (line 765-766), ungrammatical parts in a sentence and typo’s (line 21, line 246 – missing quote, line 343, line 410, line 477, line 419, line 786). In addition, left quotes are not in the right direction (for example, line 343). Certain references are not formatted correctly (line 771).
The references do provide some context, but the related work section fails to make a clear comparison of the currently proposed framework to other frameworks. It does not become clear what is concretely missing as justification for the current framework. It would be better if this section gave a clear account of the particular bias that is seen in papers applying off-the-shelf NLP, and how the papers that acknowledge these biases actually address them.

Experimental design

The experimental design is a bit opaque from the description. Three types of NLP tasks are applied on two case studies, where the generic framework is used to analyze the outcomes. There is a loose structure to this analysis: the output is analyzed in a seemingly ad hoc fashion, where certain samples of predicted messages are discussed, without any clarity about how these samples were taken. The Weather model to critically reflect on the outcomes is also rather loose in its implementation. There are many things that could be highlighted as positive or negative, and it would have been better if the different features of the approaches (speed, volume, clarity, etc.) would have been identified before filling in the different weather conditions.

Information is missing to replicate the given models. Particulary the data that models were trained on, how these data were labeled, or the way in which the number of topics was decided on for LDA are left undiscussed.

Validity of the findings

There is a lot of value in the general message of the paper, namely that NLP approaches should be applied with care and a critical reflection on their output. Unfortunately, the case studies conducted as a proof of concept do not form a good example of best practice. When applying LDA, the number of topics makes a great difference for the output, and it is totally unclear how the three topics has been decided on, nor is there an analysis whether these topics are actually clear. For sentiment analysis and emotion detection, it is important to annotate a sample of hundreds of messages as humans (including inter-annotator agreement), assessing to what extent the sentiment analysis is reliable. The authors rightfully point at negation and sarcasm as causes for a wrong score, but this is based on rather small samples of four posts and there is no clarity on how these samples were drawn.

The authors motivate the choice for a generic framework as something that is applicable to several tasks, but there is a reason why there are particular frameworks for different tasks, since the features of an approach are of importance to the way in which the outcomes can be validated. There would be a lot more gain in understanding the main workings and potential errors of an approach, and being able to analyze to what extent its predictions are useful, rather than interpreting the outcomes anyhow.

Additional comments

The suggestion is made that the current analysis is in line with explainable AI, whereas this field is characterized by an automated analysis of the model or predictions to communicate to users why a certain prediction was made. The current manual analysis with a few examples of failures is not well aligned with these endeavours.

---

## Round 0.2 · Minor Revisions

The paper should become publishable once the final minor revisions suggested by both reviewers are addressed.

Reviewer 1 ·

Basic reporting

Overall the paper is very clear. The related work section has improved, the authors added more papers that have similar goals (e.g. reflecting on computational linguistics methods).
A few small comments:

- Use ` for opening quotes (e.g. line 29)
- 101 strange sentence
- 106 that that
- 382: reference is missing parentheses
- Many figures are difficult to read when printing in black/white

Experimental design

- The paper has improved substantially in terms of how the contributions are framed and contextualized.
- line 352 mentions the programming of models with a training set. I don't know what was meant here.
Did you retrain the models? How does that work exactly with something like VADER?
- I also appreciate the added human review analyses. For clarity, these sections could be improved
by stating clearly how many tweets were annotated in total (this was not always mentioned but had to be inferred from the number of months) and how many categories were involved in the annotation.
I also wish the authors had said more about agreement on the individual categories when discussing sentiment analysis and emotion analysis results, to better understand the (somewhat) low agreement rates.

Validity of the findings

Overall findings are clear, and limitations are properly discussed.

Additional comments

Thanks for the response. Overall I think the article has improved substantially and is now ready to be published.

·

Basic reporting

Has been improved in accordance with the reviews, although in the added text the problem with the wrong left-hand quote prevailed.

Experimental design

Most significantly, a thorough procedure with manual annotations has been added, which was done in a sound way and has made the critical reflection more insightful.

The related work section has been improved, but a more structural re-writing endeavour would be at place to indicate in more detail what has been already done in terms of critical reflection on automated model output, and what knowledge gap the current study fulfills. It is still not clear what is concretely missing as justification for the current framework.

Validity of the findings

I would have liked to see a higher level discussion about the findings that the added post-review has resulted in. Particularly, seeing that humans may disagree to a significant extent about posts belonging to the same topic or a post's polarity, has implications for the text mining endeavour in general. In addition, when the human annotations indicate that there is quite some questionnable output from the off-the-shelf tool, how does that affect the analysis in steps 2 and 4? There are some shallow remarks about this, but I would expect a more in-depth discussion about both the subjectivity of a task and model performance on a given dataset, and what this implies for conclusions that can be drawn from the outcomes.

---

## Round 0.3 · accepted · Accept

I consider that the last round of revisions addressed the final minor suggestions from the reviewers and therefore I recommend acceptance of this paper in its current form.